# Extracorporeal Membrane Oxygenation for COVID 2019-Acute Respiratory Distress Syndrome: Comparison between First and Second Waves (Stage 2)

**DOI:** 10.3390/jcm10214839

**Published:** 2021-10-21

**Authors:** Nicolas Dognon, Alexandre Gaudet, Erika Parmentier-Decrucq, Sylvain Normandin, André Vincentelli, Mouhamed Moussa, Julien Poissy, Thibault Duburcq

**Affiliations:** 1Department of Intensive Care Medicine, Critical Care Centre, CHU Lille, F-59000 Lille, France; nicolas.dognon@gmail.com (N.D.); Alexandre.GAUDET@chu-lille.fr (A.G.); Erika.DECRUCQ@chru-lille.fr (E.P.-D.); normandinsylvain@gmail.com (S.N.); Julien.POISSY@chu-lille.fr (J.P.); 2Faculté de Médecine, Univ. Lille, Inserm, CNRS, CHU Lille, Institut Pasteur de Lille, U1019-UMR9017-CIIL-Centre d’Infection et d’Immunité de Lille, F-59000 Lille, France; 3Department of Cardiovascular Surgery, CHU Lille, F-59000 Lille, France; Andre.VINCENTELLI@chu-lille.fr; 4Faculté de Médecine, Univ. Lille, Inserm, Institut Pasteur de Lille, U1011—EGID, F-59000 Lille, France; Mouhamed.MOUSSA@chru-lille.fr; 5Department of Cardiovascular Intensive Care Unit, CHU Lille, F-59000 Lille, France; 6Faculté de Médecine, Univ. Lille, Inserm U1285, CNRS, UMR 8576—Unité de Glycobiologie Structurale et Fonctionnelle, F-59000 Lille, France

**Keywords:** extracorporeal membrane oxygenation, COVID 2019, acute respiratory distress syndrome, outbreak waves, respiratory drive, ventilator-associated pneumonia

## Abstract

We aimed to compare the outcomes of patients under veno-venous extracorporeal membrane oxygenation (V-V ECMO) for COVID-19-Acute Respiratory Distress Syndrome (CARDS) between the first and the second wave. From 1 March 2020 to 30 November 2020, fifty patients requiring a V-V ECMO support for CARDS were included. Patient demographics, pre-ECMO, and day one, three, and seven on-ECMO data and outcomes were collected. The 90-day mortality was 11% higher during the second wave (18/26 (69%)) compared to the first wave (14/24 (58%) (*p* = 0.423). During the second wave, all of the patients were given steroids compared to 16.7% during the first wave (*p* < 0.001). The second wave’s patients had been on non-invasive ventilation support for a longer period than in the first wave, with the median time from ICU admission to ECMO implantation being significantly higher (14 (11–20) vs. 7.7 (5–12) days; *p* < 0.001). Mechanical properties of the lung were worsened in the second wave’s CARDS patients before ECMO implantation (median static compliance 20 (16–26) vs. 29 (25–37) mL/cmH2O; *p* < 0.001) and during ECMO days one, three, and seven. More bacterial co-infections before implantation and under ECMO were documented in the second wave group. Despite a better evidence-driven critical care management, we depicted fewer encouraging outcomes during the second wave.

## 1. Introduction

Coronavirus disease 2019 (COVID-19) caused by the novel severe acute respiratory syndrome coronavirus 2 (SARS-CoV-2) can lead to severe organ dysfunction. Acute respiratory failure appears to be the main indication for critical care admission with a rate of ICU admission from 4.0% to 32% [1]. Early in the ongoing pandemic, controversy regarding ventilatory management emerged [2,3,4,5]. Some expert physiologists and several international guidelines have endorsed early intubation over an initial non-invasive strategy [6,7,8,9,10,11]. This approach was driven both by the prevention of further contamination and the lower risk of patient self-induced lung injury which has been incriminated to further worsen respiratory failure [6,7,8,9,10,11,12,13]. During the second wave of the COVID-19 outbreak, cumulative evidence influenced the global strategy of care management. An early low dose of corticosteroids has been proven to decrease the mortality in COVID-19 patients [14]. A growing body of reports revealed COVID-19-Acute Respiratory Distress Syndrome (CARDS) as being similar to classic ARDS [15,16]. Thus, physicians were less reluctant to use non-invasive ventilatory strategy. The prospective REVA network’s cohort study captured a progressive decrease in 90-day mortality with a higher proportion of patients on non-invasive ventilatory management [17]. Furthermore, other data concerning patients on invasive mechanical ventilation found an increase in mortality during the second wave [18].

In case of CARDS refractory to other management strategies, veno-venous extracorporeal membrane oxygenation (V-V ECMO) appears to provide a valuable support with conventional patient selection criteria [19,20,21,22]. The pooled estimate of prevalence of COVID-19 patients placed on ECMO was 6.4% (95% CI, 4.1–9.1) of ICU cases [23] with a cumulative incidence of 90 day-mortality at 38% (95% CI, 34.6–41.5) [20]. However, preliminary results of the European Extracorporeal Life Support Organization (ELSO) found an increase in mortality between the first and the second waves in COVID-19 patients under V-V ECMO [24]. Finally, little is known about the impact of the current bundled treatment combination (ventilatory strategies and corticosteroids) on the most critically ill COVID-19 patients on V-V ECMO for a long study period.

The aim of this study was to describe the characteristics and outcomes of patients who received V-V ECMO for CARDS between the first and the second wave of the COVID-19 outbreak.

## 2. Materials and Methods

### 2.1. Study Design and Participants

The present study was a single-center retrospective observational study. From 1 March 2020 to 30 November 2020, all consecutive adult patients with laboratory confirmed SARS-CoV-2 infection admitted to the ICU who received V-V ECMO for severe ARDS at the Lille University Hospital were included. Patients who received veno-arterial ECMO (V-A ECMO) were excluded. The period from 1 March 2020 to 3 May 2020 defines the first wave, and from 1 September 2020 to 30 November 2020 defines the second wave of the COVID-19 outbreak. All SARS-CoV-2 infections were documented by real-time RT-PCR on nasopharyngeal swabs and lower respiratory tract aspirates. Severe ARDS was defined according to Berlin’s definition [25]. Patients received V-V ECMO in case of refractory hypoxemia and/or hypercapnia despite ventilator optimization according to the ECMO to Rescue Lung Injury in Severe ARDS (EOLIA)’s criteria [26].

The French Institutional Authority for Personal Data Protection (Committees for the Protection of Human Subjects, registration no DEC21-199) approved the study. Patient data were anonymized before analysis. According to French laws, only non-opposition of the patient or their legal representative for use of the data was obtained since this observational study did not modify existing diagnostic or therapeutic strategies.

### 2.2. Data Collection and Outcome Measures

Data were collected from our electronic health records (IntelliSpace Critical Care and Anesthesia (ICCA), Philips Healthcare^®^, Böblingen, Germany).

V-V ECMO cannulation was conducted percutaneously under ultrasonography guidance by an intensivist physician or a cardiovascular surgeon. Anatomic contraindication blood drainage with a large cannula (23–29 Fr) inserted into the common femoral vein and returned through the right internal jugular vein (17–21 Fr) was recommended. The pump speed was adjusted to obtain a blood-oxygen saturation of 90% or more. The cannula position was guided by ultrasonography and verified by chest X-ray. For highly unstable patients in a secondary hospital, our mobile ECMO retrieval teams were sent to the patients’ bedside for ECMO cannulation. Once ECMO had been implanted, patients were referred to our hospital. According to preliminary reports of high thrombotic complication during management in COVID-19 patients on ECMO [19,20], systemic anticoagulation was maintained using unfractioned heparin for a targeted anti-Xa activity of 0.3–0.5 UI/mL after an initial bolus of 50–100 IU/kg. This targeted anti-Xa activity was decreased in high risk of bleeding and hemorrhagic patients. Plasma-free haemoglobin, haptoglobin and schizocytosis concentrations were monitored daily. The haemoglobin threshold for red blood cell transfusion was 7–8 g/dL (or 10 g/dL when hypoxemia persisted); platelet transfusions were discouraged except for severe thrombocytopenia (<50 × 109 cells per L) or thrombocytopenia of more than 100 × 109 cells per L with bleeding. Ultraprotective mechanical ventilation targeting lower tidal volume (1–4 mL/kg of IBW), respiratory rate (<20/min), and driving pressure (<15 cmH2O) was recommended for the first days of V-V ECMO initiation. Prone positioning under ECMO and early spontaneous breathing using airway pressure release ventilation (APRV) or spontaneous—proportional pressure support were strongly recommended.

Driving pressure (ΔP) was calculated according to Amato et al. [27], as end-inspiratory plateau pressure minus PEEP. Mechanical power (MP) was computed with surrogate formulas defined elsewhere [28,29]. Major bleeding was defined according to ELSO guidelines [19]. Massive hemolysis was defined as plasma-free hemoglobin > 500 mg/L associated with clinical signs of hemolysis. Thrombotic complications included proven pulmonary embolism and deep venous thrombosis. Reasons for circuit change included clogged circuit, thrombocytopenia, hypofibrinogenemia, acquired Willebrand’s disease, membrane lung failure (define to PaO2/FmO2 < 250), and pump failure.

The primary objective of this study was to compare the overall 90-day mortality of CARDS patients on V-V ECMO between the first and the second waves of SARS-CoV-2’s outbreak.

The secondary objective was to describe hospital mortality, ICU and hospital lengths of stay, duration of ECMO, mechanical ventilation, catecholamines and RRT, rate of ECMO weaning, prognostic scores (SAPS II [30], sequential organ-failure assessment score [31], respiratory extracorporeal membrane oxygenation survival prediction score [32]), demographic characteristics, clinical and biological parameters, respiratory support and mechanical data, adjunctive interventions, and adverse events (ischemic stroke, hemorrhagic stroke, major bleeding, thrombotic complications, massive hemolysis, circuit change, cardiac arrest, pneumothorax, ventilator associated pneumonia, bacteremia, and acute kidney injury defined as KDIGO III score) before ECMO and on ECMO at days one, three, and seven between the two phases of SARS-CoV-2’s outbreak.

### 2.3. Data Analysis

Categorical and quantitative variables were reported as percentage (%) and medians (interquartile range) or means (standard deviation) as appropriate. We compared groups using χ2 or Fisher exact tests for categorical variables, and Mann-Whitney U or Unpaired *t* tests for continuous variables. All of the analyses were computed at a two-sided α level of 5% with the software GraphPad Prism 9.1.2^®^, San Diego, CA, USA and IBM SPSS Statistics 28.0.0.0^®^, Bois-Colombes, France.

## 3. Results

From 1 March 2020 to 30 November 2020, 52 patients required a V-V ECMO assistance for CARDS. Two patients in the second wave group had a V-A ECMO support for cardiac arrest and were excluded from the analysis. The flow chart of the study is reported in Figure 1.

The main indication for ECMO implantation was a PaO2/FiO2 < 80 mmHg for >6 h concerning 38/50 (76%) patients of the global cohort with no difference between the two groups, respectively, 19/24 (79%) and 19/26 (73%) of patients. Other indications were refractory hypoxemia with PaO2/FiO2 < 50 mmHg for >3 h for two patients in each group, persistent hypercapnic acidosis with pH < 7.25 and PaCO2 > 60 mmHg for >6 h for six patients (two in the first wave and four in the second wave), and other reasons not included in EOLIA trial inclusion criteria for one patient in each group. All V-V ECMO cannulation was done percutaneously. Fourteen percent (7/50) of V-V ECMO were placed in a secondary hospital by mobile ECMO retrieval teams.

Pre-ECMO characteristics are reported in Table 1. Demographic, prognostic scores, and comorbidities between the first and second wave groups were not statistically different (except for age and SOFA). The median times from first symptoms and ICU admission to ECMO were significantly higher in the second wave group due to a significantly longer time between ICU admission and intubation and the increased use of non-invasive respiratory support before intubation. Only 2/24 patients (8%) had at least 24-h of high flow nasal oxygen or non-invasive ventilation prior to intubation during the first wave compared to 21/26 patients (81%) during the second wave, *p* < 0.0001. The median time from intubation to ECMO was similar between the first wave (7 (4–10) days) and the second wave groups (8 (3–12) days; *p* = 0.67).

Concerning biological parameters before ECMO, there was no significant difference between the two groups except for a much lower PaO2/FiO2 ratio in the second wave group (median 68 (57–75) mmHg compared to 73 (65–84) mmHg in the first wave group, *p* = 0.04), and lower inflammatory biomarkers (respectively, 7.1 (5.9–8.1) g/L for fibrinogen compared to 8 (7.2–9.3) g/L in the first wave group, *p* = 0.009 and 0.51 (0.23–1.6) ng/L for PCT compared to 1.8 (0.55–7.1) in the first wave group, *p* = 0.016). For more details, refer to Appendix A.

Details about respiratory support, adjuvant treatment, COVID-19 therapies, and complications pre-ECMO are reported in Table 2. Briefly, the whole cohort was placed on mechanical ventilation with an assist control ventilation mode before ECMO implantation. The median static compliance was significantly lower and the median driving pressure was significantly higher in second wave group. No difference was observed regarding adjuvant treatment for ARDS pre-ECMO between the two groups. In the second wave group, all patients received glucocorticoids with a median time to ECMO of 13 (11–19) days while only 16.7% patients of the first wave group with a median time to ECMO of 7.5 (5.3–9) days (*p* = 0.015). Antiviral treatments were significantly more regularly administered in the first wave group in comparison to the second wave group. Regarding complication before ECMO implantation, we reported much higher pneumothorax and secondary bacterial infection in the second wave group. These infections were mainly documented bacterial ventilator associated pneumonia with respectively 5/24 (21%) in the first wave group and 13/26 (50%) in the second wave group (*p* = 0.032). No difference regarding bacteremia was reported between the two wave groups, respectively, 3/24 (12.5%) and 5/26 (19.2%).

ECMO, ventilation, biological parameters, and SOFA score at day one, three, and seven under V-V ECMO between first wave and second wave groups are reported in Table 3, Appendix A. The SOFA score was not statistically different between the first (12 (10–14)) and the second wave groups (11 (9.5–13); *p* = 0.07) at day one under V-V ECMO. During the ECMO course, 44/50 (88%) patients received glucocorticoids, respectively, 22/24 (91.7%) in the first wave group and 22/26 (84.6%) in second wave group, *p* = 0.669. The inflammatory biomarkers remained lower at ECMO day one, three, and seven in the second wave group.

Concerning ARDS adjuvant treatments, 34/50 (68%) patients were prone positioned in the whole cohort, with 13/24 (54.2%) in the first wave group and 21/26 (80.8%) in the second wave group (*p* = 0.044). No significant difference between groups was noted concerning the use of inhaled nitric oxide and almitrine. Ninety-eight percent patients received red cells transfusion under ECMO with median 8.5 (5–14) packed red blood cells, 34% received platelets, 28% received fresh frozen plasma, and 14% received fibrinogen concentrate under ECMO with no difference between the two groups. For more details, refer to Appendix A.

Outcomes and complications under ECMO are reported in Table 4. The overall 90-day mortality of COVID-19 patients under V-V ECMO was 32/50 (64%) in our cohort. This mortality rate was higher during the second wave 18/26 (69%) compared to first wave 14/24 (58%), but without reaching statistical significance (*p* = 0.423). Renal replacement therapy was used during ECMO support in nearly half of the whole cohort (22 (44%) patients) and significantly more frequently during the first wave.

## 4. Discussion

We report here a retrospective single institution study regarding the impact of the current bundled treatment combination (ventilatory setting management and corticosteroids) on the most critically ill COVID-19 patients under V-V ECMO for a long study period. The main finding of our study was a 11% higher 90-day mortality during the second wave, albeit without reaching statistical significance due to the small size of the cohort.

The overall 90-day mortality rate of CARDS patients on V-V ECMO in our study was 32/50 (64%). Data from high-volume centers with retrospective design show that ECMO therapy was associated with a lower in-hospital mortality rate ranging from 36 to 54% [33,34,35]. Data from the international ELSO Registry captures a 52.4% in-hospital mortality in 1531 treated patients as of September 14th 2020 [36]. Barbaro et al. reported in the subset of CARDS patients receiving V-V ECMO an estimated cumulative incidence of in-hospital mortality 90 days after the initiation of ECMO of 38.0% (95% CI 34.6–41.5) [20]. A recent multinational meta-analysis confirmed a lower pooled in-hospital mortality of 37.1% (95% CI 32.3–42.0%) of COVID-19 patients receiving ECMO (22 studies, 1896 patients) [37]. However, all of these studies took place solely during the first wave of COVID-19’s pandemic. In contrast, other studies reported a comparable high mortality such as a recent multicenter study in Germany involving a total of 768 COVID-19 ECMO patients admitted to hospitals between February and December 2020 with a 73% in-hospital mortality [38]. The reasons for the high in-hospital mortality in our cohort might be the higher mean age of 58 (±10) years in the whole cohort. This is comparable to the mean age of 57.7 (±11.4) years in the study by Karagiannidis et al. [38] but is significantly higher than previous studies with a mean age ranging from 48 (±11) to 55.4 (±9.3) years [20,34,35,36]. Increasing age is one of important pre-ECMO variables associated with a worse outcome, as demonstrated by many studies [20,34,38,39]. Another factor, contributing to the higher mortality rates in our cohort, may be the median SAPSII (58 (34–67)) higher than the ones reported by Schmidt et al. [35] (median 45 (29–56)) and by Lebreton et al. (median 40 (31–56)) [34]. Although not characterized to COVID-19 ARDS, the RESP score [32] reported by Schmidt et al. [35] was 4 (2–5) and 3 (1–5) by Diaz et al. [33], which is significantly higher than in our cohort with a median of −3 (−6,−1) and an estimated survival probability of 33%. The overall 90-day mortality rate of COVID-19 ICU patients was 116/502 (23.1%) in the whole cohort of our study with no significant statistical difference between the two waves (22% vs. 24%) regarding ICU mortality rates by age.

Regarding specific COVID-19 therapies during the second wave, several treatment options have been established since then, such as the systematic use of corticosteroids in our second wave patients’ group in light of the RECOVERY trial [14] rather than only four (16.7%) patients during the late first wave. Due to extent inclusion in international clinical trials, antiviral treatments were significantly more administered in the first wave group rather than in the second wave group (respectively, 11/24 (45.8%) and 3/26 (11.5%); *p* = 0.007)), in light of recent trials demonstrating that none of these treatments are efficient in treating COVID-19 [40,41], including patients requiring mechanical ventilation [42,43,44]. Regarding non-invasive respiratory support strategies, there is a paucity of high-quality evidence in COVID-19 resulting in marked variation in international practice [45]. Interestingly, a recent report providing data from the EuroELSO survey indicate a trend to less favorable outcomes during the second wave. Including deaths reported after successful weaning, survival was 53% in the first wave and 44% in the second wave (*p* < 0.0001) [24]. In the same way, we found an 11% higher 90-day mortality during the second wave in our study. Several hypotheses might be considered.

First, the second wave patients were significantly older than those from the first wave (*p* < 0.017). As already discussed previously, age is a well-established risk factor for worse outcomes. Comorbidities were not different between the two wave groups and similar to other large cohort studies [20,33,34,35,36,38]. Otherwise, the patients during the second wave were prone to manifest higher severity of pulmonary involvement even if they had less extra-pulmonary organ dysfunction. Indeed, we reported a trend towards a lower SAPSII score, a statistically significant lower total SOFA score, and a trend towards a worsened RESP score in the second wave group. We may speculate that the more frequent use of corticosteroids during the second wave had mitigated the cytokine release syndrome observed in severe COVID 19 [46] and the extrapulmonary organ dysfunctions.

Secondly, the longer delay between ICU admission and intubation due to a more frequent use of non-invasive respiratory support during the second wave was responsible for a longer time from ICU admission to ECMO. This could contribute to the worsened respiratory phenotype at ECMO implantation. Indeed, the static compliance of the respiratory system was significantly lower with a significantly higher driving pressure in the second wave group. Biological parameters corroborate this data with a significantly lower PaO2/FiO2 ratio than in the first wave group and a trend to hypercapnic acidosis with an equilibrated pH due to significantly higher alkaline reserve. To note, no significant difference in adjuvant treatment (prone positioning, neuromuscular blockade, inhaled nitric oxide) before ECMO implantation was noticed between the two groups. Thus, the more severe respiratory phenotype during the second wave may be due to (1) an ECMO initiation at a more advanced stage of the disease, and (2) patient self-inflicted lung injury secondary to vigorous respiratory drive during non-invasive support as already described in other reports [3,13]. As such, we observe much more pneumothorax before ECMO canulation in the second wave group. Finally, the same trend towards a more severe respiratory phenotype persisted under ECMO at day one, three, and seven in the second wave group.

Thirdly, the relative immunodepression related to systematic use of corticosteroid, as could be suggested by the lymphopenia being significantly lower on ECMO day seven during the second wave, might contribute to less favorable outcomes. On the one hand, we documented significantly more bacterial co-infection before ECMO implantation in our study with 61.5% patients during the second wave period compared to 21% in the first wave period (*p* = 0.004) with a predominance of bacterial ventilator-associated pneumonia, which could contribute to the worsened respiratory phenotype at ECMO implantation. On the other hand, we reported much higher antibiotic blood stream infections under ECMO during the second wave, that could partly explain the higher length of ECMO, catecholamines, and ICU and hospital stay.

Several limits must be highlighted in our study. First, the limited size of our cohort and the retrospective and monocentric design which exposed the study to confounders may have resulted in underpowered analyses. Nevertheless, as all ECMO patients were referred to our tertiary center, patient management during the two periods of the study were homogenous and allowed to make relevant comparison. Moreover, the changes in patient care between the two waves were especially evidence-driven and thus not preclude to extent these results to other centers [47]. Second, the statistically significant difference of age during the second wave group is a major limitation and assumptions can only be made about the meaning of other pre-ECMO variables associated with a worse outcome during the second wave. Data from larger multicentric comparative studies are needed to confirm this trend. Third, the responsibility of circulating virus strains between the first and second wave seems unlikely, since the original European SARS-CoV-2 represented the main strain circulating in France until mi-January 2021. Fourth, the different survival rate was not related to a selection bias due to a greater workload on the national health system. In fact, we observed the same burden during the two waves in our tertiary center and have not encountered any problems in terms of availability for ECMO’s material.

## 5. Conclusions

Our study describes the impact of the current bundled treatment combination (ventilatory setting management and corticosteroids) on the most critically ill COVID-19 patients on V-V ECMO during the two waves of COVID-19 outbreak and reveals that the clinical picture is less encouraging during the second wave with a trend of an increase in 90-day mortality. Further data analysis is expected to support our preliminary results and provide valuable data on ECMO management strategies for these patients.

## Figures and Tables

**Figure 1 jcm-10-04839-f001:**
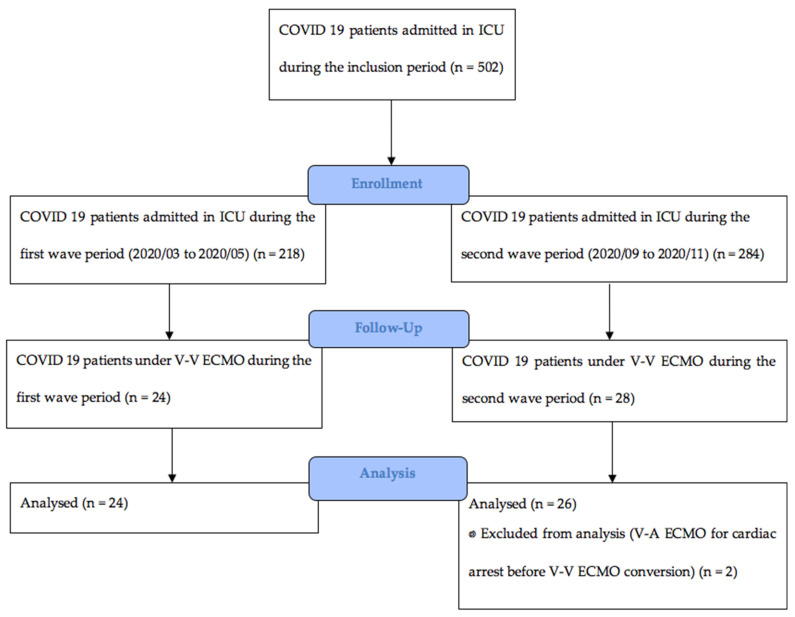
Flow chart of the study. V-V ECMO: veno-venous extracorporeal membrane oxygenation. V-A ECMO: veno-arterial extracorporeal membrane oxygenation.

**Table 1 jcm-10-04839-t001:** Patients’ characteristics at veno-venous extracorporeal membrane oxygenation (V-V ECMO) initiation in the first and second wave groups.

Characteristics at ECMO Initiation	All Patients (=50)	First Wave (=24)	Second Wave (=26)	*p*-Value
Age (years)	61 (53–66)	58 (49–63)	63 (59–67)	0.017
Male	46 (92)	21 (87.5)	25 (96.2)	0.34
BMI (kg/m²)	31 (28–36)	33 (29–38)	30 (27–35)	0.158
SAPSII	58 (34–67)	60 (42–69)	42 (32–67)	0.079
SOFA	10 (8–12)	11 (9–12)	9 (8–11)	0.027
RESP	−3 (−6,−1)	−2 (−4,−1)	−5 (−6.3,−1)	0.063
No Comorbidities	9 (18)	6 (25)	3 (11.5)	0.216
HTA	27 (54)	11 (46)	16 (61.5)	0.266
Diabetes	18 (36)	9 (37.5)	9 (34.6)	0.832
Dyslipidemia	20 (40)	7 (29.2)	13 (50)	0.133
Obesity (BMI > 30)	29 (58)	16 (66.7)	13 (50)	0.233
Malignancy	1 (2)	1 (4)	0 (0)	0.48
Other immunocompromised condition	6 (12)	2 (8.3)	4 (15.4)	0.669
Time from first symptoms to ECMO (days)	16 (14–26)	15 (11–16)	19 (16–26)	0.004
Time from first symptoms to ICU (days)	7 (5–9)	7 (5–9)	6 (4–9)	0.33
Time from ICU admission to ECMO (days)	12 (6–15)	7.5 (5–12)	14 (11–20)	<0.001
Time from ICU admission to intubation (days)	1 (0–6)	0 (0–1)	5,5 (1–9)	<0.001

Data are median (IQR) or *n* (%). BMI = body mass index. SAPSII, Simplified Acute Physiology Score II; SOFA, Sequential Organ-Function Assessment; RESP, Respiratory Extracorporeal Membrane Oxygenation Survival Prediction; ECMO, extracorporeal membrane oxygenation.

**Table 2 jcm-10-04839-t002:** Respiratory support, adjuvant treatments, COVID-19 therapies, and complications pre-ECMO in the first and second wave groups.

Pre-ECMO Characteristics	All Patients (=50)	First Wave (=24)	Second Wave (=26)	*p*-Value
**Respiratory Support**				
FIO2	100 (100–100)	100 (100–100)	100 (88–100)	0.082
Vt (mL)	420 (380–460)	425 (393–480)	410 (380–440)	0.08
Vt IBW (mL/kg)	6.1 (5.7–6.6)	6.5 (5.7–7)	6 (5.3–6.2)	0.009
RR (bpm)	30 (26–31)	30 (26–30)	30 (30–32)	0.159
Ppeak (cmH2O)	40 (36–45)	43 (38–47)	38 (35–42)	0.068
Pplat (cmH2O)	30 (28–32)	30 (27–32)	31 (30–32)	0.402
PEEP (cmH2O)	12 (7.5–15)	14 (12–16)	10 (5–14)	<0.001
Driving Pressure (cmH2O)	17 (14–22)	15 (12–17)	21 (17–24)	<0.001
Static Compliance (mL/cm H2O)	25 (18–29)	29 (25–37)	20 (16–26)	<0.001
Mechanical Power (J/min)	35 (30–47)	43 (34–52)	32 (28–39)	0.004
**Adjuvant treatment**				
Prone Positioning (PP)	48 (96)	24 (100)	24 (92)	0.491
Number of PP before ECMO	3 (2–5)	3 (2–5)	3 (2–5)	0.979
Neuromuscular Blockade	50 (100)	24 (100)	26 (100)	1
Inhaled nitric oxide	44 (88)	21 (87.5)	23 (88.5)	1
Almitrine	29 (58)	11 (45.8)	18 (69)	0.094
**COVID-19 therapies**				
Glucocorticoids	30 (60)	4 (16.7)	26 (100)	<0.001
Antiviral	14 (28)	11 (45.8)	3 (11.5)	0.007
**Complications pre-ECMO**				
Renal replacement therapy	8 (16)	5 (21)	3 (11.5)	0.456
Pulmonary embolism	15 (30)	7 (29)	8 (31)	0.902
Pneumothorax	5 (10)	0 (0)	5 (19.2)	0.051
Documented bacterial co-infection	21 (42)	5 (21)	16 (61.5)	0.004

Data are median (IQR) or *n* (%). Antiviral therapies were Lopinavir-Ritonavir, Chloroquine, Remdesivir. FiO2, fraction of inspired oxygen; Vt = tidal volume; Vt IBW, ideal body weight tidal volume; RR, respiratory rate; Ppeak, peak pressure; Pplat, plateau pressure; PEEP, positive end-expiratory pressure.

**Table 3 jcm-10-04839-t003:** ECMO, ventilation, and biological parameters at veno-venous extracorporeal membrane oxygenation (V-V ECMO) day one in the first and second wave groups.

Day 1 Characteristics	Parameters	All Patients (=50)	First Wave Group (=24)	Second Wave Group (=26)	*p*-Value
**ECMO parameters**	FmO2 (%) ^α^	100 (80–100)	100 (80–100)	95 (79–100)	0.648
	RPM ^β^	3600 (3300–4033)	3500 (3200–4065)	3600 (3375–4025)	0.705
	ECMO blood flow (L/min) ^γ^	5.5 (5–6)	5.9 (5.5–6.1)	5.1 (4.8–5.5)	0.025
	Sweep gaz flow (L/min) ^δ^	5 (4.3–6)	6 (4–6)	5 (4.4–6.3)	0.336
**Ventilation parameters**	FiO2 (%) ^φ^	50 (40–50)	50 (40–60)	50 (40–60)	0.674
	Vt (mL) ^χ^	250 (180–295)	280 (240–300)	230 (180–250)	0.197
	Vt IBW (mL/kg) ^ε^	3.6 (2.7–4.3)	4 (3.5–4.8)	3.4 (2.4–3.9)	0.223
	RR (cpm) †	17 (13–20)	16 (14–20)	17 (12–21)	0.82
	Pplat (cmH2O) #	24 (20–26)	25 (22–27)	23 (20–25)	0.034
	PEEP (cmH2O) ‡	12 (10–14)	14 (10–16)	10 (10–12)	0.007
	Driving Pressure (cmH2O) ¶	12 (10–14)	11 (10–15)	12 (10–14)	0.376
	Compliance RS (mL/cm H2O) ¥	21 (14–30)	23 (17–31)	19 (12–26)	0.273
	Mechanical Power (J/min) ¤	9.4 (6.6–15)	12 (9–17)	7.4 (4.4–10)	0.01
**Biological parameters**	pH	7.4 (7.3–7.5)	7.4 (7.3–7.5)	7.4 (7.3–7.5)	0.836
	PaO2 (mmHg)	75 (65–85)	76 (67–87)	73 (65–84)	0.299
	PaCO2 (mmHg)	47 (42–55)	44 (40–50)	52 (44–59)	0.016
	Bicarbonates (mmol/L)	30 (26–34)	28 (24–32)	30 (27–35)	0.089
	Hemoglobin (g/dL)	8.7 (7.9–9.8)	8.5 (8–9.7)	8.7 (7.6–10)	0.841
	Platelets (10^9^/L)	234 (162–301)	245 (178–326)	202 (156–267)	0.163
	Fibrinogen (g/L)	7.1 (5.4–8.2)	7.9 (6.6–8.7)	6.1 (4.6–7.6)	0.002
	aPTT (ratio)	1.5 (1.2–1.8)	1.5 (1.2–1.8)	1.5 (1.2–1.7)	0.662
	CRP (mg/L) ^d^	148 (92–252)	178 (104–329)	107 (79–176)	0.025
	PCT (ng/mL) ^e^	1 (0.3–3.2)	2.3 (0.6–4.8)	0.48 (0.2–1.8)	0.048

Values are number (%) or median (interquartile range). FmO2, fraction of membrane oxygen; RPM, rate per minute; FiO2, fraction of inspired oxygen; Vt, tidal volume; Vt IBW, ideal body weight tidal volume; RR, respiratory rate; Ppeak, peak pressure; Pplat, plateau pressure; PEEP, positive end-expiratory pressure; Compliance RS, respiratory system compliance; aPTT, activated partial thromboplastin time; ASAT, aspartate aminotransferase; ALAT, alanin aminotransferase. ^α^ 1 missing value in first wave group, ^β^ 1 missing value in first wave group, ^γ^ 1 missing value in first wave group, ^δ^ 1 missing value in first wave group, ^φ^ 1 missing value in first wave group, ^χ^ 1 missing value in first wave group, 1 missing value in second wave group, ^ε^ 1 missing value in second wave group, † 1 missing value in first wave group, 1 missing value in second wave group, # 1 missing value in first wave group, 1 missing value in second wave group, ‡ 1 missing value in first wave group, 1 missing value in second wave group, ¶ 1 missing value in first wave group, 1 missing value in second wave group, ¥ 1 missing value in first wave group, 1 missing value in second wave group, ¤ 1 missing value in first wave group, 1 missing value in second wave group, ^d^ 1 missing value in first wave group, ^e^ 2 missing values in first wave group.

**Table 4 jcm-10-04839-t004:** Outcomes and complications under ECMO in First and Second Wave Groups.

Outcomes and Complications under ECMO	All Patients (=50)	First Wave (=24)	Second Wave (=26)	*p*-Value
**Outcomes**				
Lenght of stay ICU (days)	33 (20–60)	25 (17–42)	35 (26–71)	0.055
Lenght of stay Hospital (days)	33 (21–64)	26 (19–53)	40 (26–97)	0.021
Length of Catecholamines (days)	14 (7–17)	8 (6–16)	15 (9–27)	0.049
Length of RRT (days)	0 (0–10)	5 (0–13)	0 (0–6.3)	0.045
Length of Mechanical ventilation (days)	23 (16–45)	21 (15–38)	29 (19–61)	0.097
ECMO weaning	20 (40)	11 (46)	9 (35)	0.419
ECMO duration (days)	12 (7–16)	11 (6–13)	14 (8.8–25)	0.013
Tracheotomy	16 (32)	6 (25)	10 (38)	0.308
Hospital mortality	31 (62)	14 (58)	17 (65)	0.608
**Complications**				
Ischemic stroke	2 (4)	2 (8.3)	0 (0)	0.225
Hemorrhagic stroke	6 (12)	4 (16.6)	2 (7.7)	0.409
RRT	22 (44)	15 (62.5)	7 (26.9)	0.011
Hemorrhagic—Site canulation	32 (64)	12 (50)	20 (76.9)	0.048
Hemorrhagic—Other	33 (66)	14 (58.3)	19 (73.1)	0.272
Thrombotic	8 (16)	5 (20.8)	3 (11.5)	0.456
Circuit change	20 (40)	8 (33.3)	12 (46.2)	0.355
Massive Hemolysis	11 (22.9)	8 (33.3)	3 (12.5)	0.086
Cardiac arrest	3 (6)	2 (8.3)	1 (3.8)	0.602
Pneumothorax	7 (14)	3 (12.5)	4 (15.4)	1
Antibiotic-treated blood stream infection	30 (60)	11 (45.8)	19 (73)	0.049
Antibiotic-treated VAP	33 (66)	15 (62.5)	18 (69.2)	0.616

Values are number (%) or median (interquartile range). RRT, renal replacement therapy; VAP, ventilator associated pneumonia. Other sites of hemorrhage: urinary tract, pulmonary tract, gastrointestinal tract, ear, nose and throat.

## Data Availability

Details regarding data supporting reported results can be submitted by the corresponding author.

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
