# Peer review of "Extracorporeal Membrane Oxygenation for COVID 2019-Acute Respiratory Distress Syndrome: Comparison between First and Second Waves (Stage 2)"

_jcm, 2021, doi:10.3390/jcm10214839_

Round 1
Reviewer 1 Report
With great interest I read the manuscript "Extracorporeal Membrane Oxygenation for COVID 2019-acute respiratory distress syndrome: Comparison between first and second waves".
It is extremely well investigated, worked off and written.
Author Response
Dear Reviewer,
Please find attached our author's reply to the Review Report
With kind regards

Reviewer 2 Report
The authors have undertaken a study to compare ECMO outcomes between two waves of the SARS-CoV-2 pandemic. The finding of increased mortality during the second wave is an interesting finding, and the hypotheses generated to explain this finding seem reasonable. The meaningfulness of the data however, is limited due to the weaknesses already discussed by the authors (retrospective single center design, small size, residual confounding etc).
In addition to demographics, and treatment factors, were there differences in circulating virus strains between the first and second wave?
Author Response

(The authors gave the same response as above.)

Reviewer 3 Report
This paper analyze differences in V-V ECMO patients between 1th and 2nd waves in a single tertiary center. I appreciated the precise analysis of the ventilation mode during ECMO run. Regarding discussion, I have some suggestions. I think it would be necessary to point out the different "workload" on the national health system between two waves: don't you think that different survival rate is related to a selection bias (was patients' access to advanced care like ECMO the same in the first wave?)? I think it would be very interesting to discuss mortality rate between similar age groups in the two waves, and not only data regarding ECMO patients. Moreover, please use CONSORT flowchart for figure 1 (http://www.consort-statement.org/consort-statement/flow-diagram)
Author Response

(The authors gave the same response as above.)

Round 2
Reviewer 3 Report
Paper was properly revised